# Evaluation of a Program to Reduce Home Environment Risks for Children with Asthma Residing in Urban Areas

**DOI:** 10.3390/ijerph19010172

**Published:** 2021-12-24

**Authors:** Brandon Workman, Andrew F. Beck, Nicholas C. Newman, Laura Nabors

**Affiliations:** 1Department of Environmental and Public Health Sciences, University of Cincinnati, Cincinnati, OH 45267-0056, USA; Nicholas.Newman@cchmc.org; 2Cincinnati Children’s Hospital Medical Center, Department of Pediatrics, College of Medicine, University of Cincinnati, Cincinnati, OH 45229-3026, USA; Andrew.Beck1@cchmc.org; 3Department of Health Promotion and Education, University of Cincinnati, Cincinnati, OH 45221-0068, USA; Laura.Nabors@uc.edu

**Keywords:** asthma, home environment, community programs, public health education, program evaluation

## Abstract

Pediatric asthma morbidity is often linked to challenges including poor housing quality, inability to access proper medical care, lack of medications, and poor adherence to medical regimens. Such factors also propagate known disparities, by race and income, in asthma-related outcomes. Multimodal home visits have an established evidence base in support of their use to improve such outcomes. The Collaboration to Lessen Environmental Asthma Risks (CLEAR) is a partnership between the Cincinnati Children’s Hospital Medical Center and the local health department which carries out home visits to provide healthy homes education and write orders for remediation should code violations and environmental asthma triggers be identified. To assess the strengths and weaknesses of the program, we obtained qualitative feedback from health professionals and mothers of children recently hospitalized with asthma using key informant interviews. Health professionals viewed the program as a positive support system for families and highlighted the potential benefit of education on home asthma triggers and connecting families with services for home improvements. Mothers report working to correct asthma triggers in the home based on the education they received during the course of their child’s recent illness. Some mothers indicated mistrust of the health department staff completing home visits, indicating a further need for research to identify the sources of this mistrust. Overall, the interviews provided insights into successful areas of the program and areas for program improvement.

## 1. Introduction

Asthma is among the most common chronic diseases in children [1]. Asthma affects approximately seven million children in the United States (U.S.) with annual asthma-related healthcare expenditures of USD 56 billion. This high cost is primarily associated with emergency department (ED) visits and hospitalizations [2]. Fifty-four percent of children with asthma <18 years reported having one or more asthma attacks in the preceding 12 months, with 18.3 per 10,000 hospitalized for asthma [3]. Utilization of acute services for asthma complications occurs at disproportionately high rates among minority and low-income children living in more urban areas [2]. Black children specifically experience nearly twice as many ED visits and hospitalizations for asthma, and experience four times more asthma-related deaths when compared to White children [4].

Over the past three decades, the greatest increases in asthma prevalence have been in urban areas and have disproportionately affected youth in racial and ethnic minority groups and those living in poverty [5]. Such at-risk populations are also affected by diminished access to primary and subspecialty care—i.e., those of minority race or ethnicity in the U.S. are less likely to be referred to asthma specialists [6]. Lacking a primary care provider to coordinate medical care and asthma management is just one indication of disparities in asthma care among racial and ethnic minority groups [6]. Additionally, communities of color and the poor experience disproportionate environmental exposures and related increased asthma symptoms, including those rooted in substandard housing conditions [7].

Housing in urban environments, and substandard housing in particular, is associated with indoor allergenic triggers, such as excessive moisture which allows for the breeding of mice, cockroaches, and dust mite allergens [8]. Moisture within the home is also associated with mold and volatile organic compounds that have been linked to increased respiratory tract infections, asthma prevalence, and asthma morbidity [9]. There is a strong evidence base for addressing environmental triggers as part of comprehensive asthma management [10]. Home visits that incorporate parental education are an effective intervention to improve asthma control that helps families reduce exposure to multiple indoor asthma triggers [11,12]. Because a healthy home environment is critical to successful asthma management, it remains important to examine ways in which the varying components of asthma intervention programs can be improved to best serve parents of children with asthma in urban areas.

This study focused on an evaluation of an asthma home environmental intervention program, Collaboration to Lessen Environmental Asthma Risks (CLEAR). CLEAR serves children hospitalized for asthma exacerbations in Cincinnati, Ohio, a demographic that is predominantly of minority race and low income [13]. This program’s referrals target children admitted to Cincinnati Children’s Hospital Medical Center (CCHMC) for asthma complications who reside within the jurisdiction of the Cincinnati Health Department (CHD), and who have reported cockroaches, rodents, mold/mildew, water damage, and/or cracks in the walls or ceilings within their homes. Hospital staff identify patients who are eligible for a referral to CLEAR by use of standardized asthma-specific history and physical and/or via a separate questionnaire administered during their hospitalization, known as the Child Asthma Risk Assessment Tool (CARAT) [14]. Parents or caregivers who accept a referral receive a telephone call from a health department sanitarian to schedule a home inspection to identify household asthma triggers. The sanitarian’s findings could prompt a written work order to be issued to the property’s landlord (or to the tenant) for the abatement of the identified hazards should violations of city ordinances be identified [13].

Much like other asthma intervention programs, CLEAR faced difficulties in reaching and enrolling eligible families, completing home assessments, and reaching participants for follow-up activities [13]. In the initial implementation of the CLEAR program, there were 99 eligible patients who received a referral to the program, 75 accepted referrals during the inpatient stay. Of the 24 who declined, 3 reported the issue was already being addressed and 2 had plans to move; 19 had no documented reason for refusal. The 75 referrals resulted in 50 inspections; reasons for non-visits included an inability to reach the family (*n* = 15) or a family that had changed their mind (*n* = 8) [13]. Considering the significance of multifaceted home intervention programs to combat asthma morbidity among children, it remains critical to determine how to reach more families with program services. We, therefore, undertook this qualitative evaluation of the CLEAR program to identify stakeholder perceptions of successes, areas for improvement, and to promote the program’s sustainability and future development.

## 2. Materials and Methods

### 2.1. Procedures

Researchers conducted key informant interviews between February and May of 2020. Interviews were completed via telephone with the exception of one in-person interview. Prior to interviews, participants received an information and informed consent form either electronically via email or by a letter explaining the purpose of the project. Three attempts were made to contact participants. Participants verbally agreed to participate in interviews and some granted the interviewer permission to audio record their interviews. Audio recordings were then deleted after an interview transcript was created for each interview.

This project was conducted in accordance with the ethical standards of the University of Cincinnati Institutional Review Board.

### 2.2. Participants

Participants for our convenience sample were referred by program leadership at the CCHMC and the CHD. The email addresses of health professionals (*n* = 10) who had provided services in the program or referred patients were provided by the children’s hospital. Parents (*n* = 37) were referred by professionals at the health department by way of a list that included the names, mailing addresses, a primary phone number, and a secondary phone number for each parent who had received services from CLEAR in the year 2019.

### 2.3. Assessment and Measures

Interview questions for health professionals were designed to collect participants’ perceptions of effective components of an environmental asthma intervention program and areas for improvement. Questions were modified from a cross siteevaluation of a pediatric asthma project, examining five asthma intervention programs inthe U.S. (New York City, New York; San Juan, Puerto Rico; Chicago, Illinois; Los Angeles, California; and Philadelphia, Pennsylvania), known as the Merck Childhood Asthma Network, Inc. [15]. Health professionals completed an interview, lasting approximately 15 min, addressing the following questions: How is the program promoting positive behaviors toward asthma among affected individuals? (Probes: Do you know about the CLEAR program and/or can you provide any information to me about this program? Explain what you mean; what services, which behaviors? How are health inequities addressed?)How is the program being implemented? (Probes: Detail program successes and areas for improvement; ask for examples and specifics/details)What are successful strategies for sustainability in services provided by the home environment intervention program? (Probes: Please provide an example; please add more information)What are challenges for sustainability in each service provided by CLEAR and/or environmental interventions from Children’s Hospital? (Probes: Please provide an example; please add more information)What are successful interventions of the CLEAR program? Why? (Probes: Please provide an example, can you explain? Are any interventions addressing health disparities? If yes, how?)Which interventions of the CLEAR program are difficult to implement? Why? (Probes: What could be altered or adjusted to make these interventions more easily implemented?)In your opinion what are the challenges in persuading families to accept and follow through with the referral process of the program? (Probes: Please provide an example; please add more information)Are there any budget constraints or financial barriers that inhibit CLEAR from operating and/or expanding services? (Probes: Ask for a specific example and further details)

Mothers who were reached by telephone typically asked for a shortened interview with fewer questions, due to time constraints (i.e., they indicated that they had limited availability to talk to researchers), resulting in 5 min interviews. Thus, interviewers first provided introductions and explained the study was designed to evaluate the CLEAR program and then asked the following questions: “Was the health department able to connect with you? Do you have an idea of what the triggers are in your home? Have you tried to correct triggers in your home yourself? Does your child’s asthma worsen seasonally?”

Following interviews, four trained, independent coders examined the responses of health professionals and parents over a series of four, one-hour meetings. In the first meeting, an open coding approach was used to determine relevant categories that emerged in the data representing health professional responses [16,17]. Through a constant comparative process over the next two meetings, coders determined a list of themes in health professional responses to different interview questions and developed a list of representative quotes for each theme. Themes and representative quotes were then organized into tables. Coders also reviewed themes and determined the meanings of descriptive information in mothers’ responses to interview questions. Disagreements were resolved by consensus. At the fourth meeting, the research team reviewed the presentation of results in the tables, and themes and representative quotes were finalized and agreed upon.

## 3. Results

Health professionals, including nine personnel from the children’s hospital and one sanitarian from the CHD completed interviews. Professionals from the children’s hospital included one respiratory therapist, one social worker, and seven pediatric physicians who see patients in the inpatient units that refer patients to CLEAR. There were nine mothers who completed interviews, eight were African American and one was Caucasian. All resided in low-income neighborhoods.

*Perceptions from Health Professionals*. Health professionals’ ideas for program sustainability, ideas for improving the program, perceptions of successful interventions, and ideas for promoting parent adherence to recommendations are presented in Table 1, Table 2 and Table 3. Health professionals viewed the referral process as positive, and sustainable (see Table 1). For example, the CARAT was viewed as helpful in identifying the need for referral. One resident physician said,
“The CLEAR program really focuses on those patients and families that have an environmental risk like rodents or roaches, or mold and mildew, you know things that aren’t up to code and I think that the way we screen, at least in the inpatient setting for the social determinants can be really enlightening for the residents in terms of knowing more about their patients and their families and what they are being affected by at any given time.”

One challenge was getting accurate information about home environments from parents to know when to initiate a referral, while another will be to improve communication between the health department and parents so that referrals occur quickly and efficiently assist parents in addressing environmental triggers. A lack of feedback from the health department about the results of the home visits made it difficult for the hospital staff to know if the program was working successfully for families (Table 1). Professionals indicated that establishing an open line of communication with the health department to report which patients were receiving home visits and services from the program would be a successful strategy for sustainability. Health professionals also highlighted communication limitations between parents and the health department that prevent sanitarians from completing home visits and environmental interventions (Table 1). Various challenges arise relating to reaching families once they leave the hospital. Contact information shared with health department staff is not always correct, which prevents them from scheduling home visits. For example, parents’ landline or cellular telephones may be disconnected or their voicemail boxes may be full. Parents may also move, or plan to move, to a new home or apartment building after receiving information from health professionals but before the sanitarian can complete a visit, particularly if they believe that moving would improve environmental conditions related to their child’s asthma. Health professionals also believed that parents may fear that their child will be removed from parental care if the health department inspects or visits the home (Table 1). To illustrate, one physician, a fellow, mentioned,

“I think that there’s a variety of other reasons why families might be hesitant to engage with the CLEAR program or home health or other programs that require outreach to them in their home setting… They are often very hesitant to provide information about where they live, give correct information about where they live, and follow up with people even via phone, let alone having someone come to their home.”

Table 2 provides information on health professionals’ views of program successes, and participants agreed that the CLEAR program was a mechanism for addressing health disparities related to conditions in the home environment. Health professionals also viewed the program as necessary for providing parents and guardians with education on identifying household triggers for asthma. They saw the program as being effective in connecting parents to the health department for resources for identifying and abating environmental triggers in the home that were affecting children’s symptoms. Positive patient feedback was identified as the best strategy for continued program support among hospital personnel.

Despite the challenges associated with connecting families with the health department, the program was viewed by health professionals as having the potential to provide a layer of support for families to get hazards within their home corrected. Health professionals considered some families to be afraid to approach their landlords with issues related to poor environmental conditions out of fear of eviction, and advocating for families was viewed as a successful intervention of the program among health professionals (Table 2). A referral to the CLEAR program was viewed as a mechanism indicating a need for educating families on the identification of environmental triggers, such as mold and pests, and having these triggers abated from the home (Table 2). The sanitarian acknowledged that this was the primary function of home visits, reporting,

“Well when you have the situation whether it be the roaches, the mold, the mice, situations abated and removed from the living area, the apartment, the townhome… I am assuming that it helps with the asthma conditions of the child, but as far as us, anytime we can get abatement is a huge success for us.”

Table 3 provides information about health professionals’ views of challenges for parents, which limit adherence to the referral process. Professionals provided information indicating the program is not reaching some families because they do not feel comfortable reporting that they have an issue in their home. This information was consistent with earlier answers suggesting that parents were fearful of having the health department in their homes (e.g., fearful of eviction, worried over confronting landlords regarding environmental issues). Professionals viewed parents as being more hesitant to engage with sanitarians for a home inspection if there were issues present in the home that could be affecting their child’s health. For instance, a pediatric resident said, “We might be missing CLEAR referrals if families don’t feel comfortable reporting that they have issues.”

Perceptions from Mothers. Mothers indicated they did receive basic information for identifying environmental triggers in their home during their child’s hospitalization. They were taking action to help reduce the impact of triggers for their child in several ways: (1) by deep cleaning the home, (2) purchasing new bedding for their child, (3) spraying for roaches, (4) altering smoking habits (e.g., smoking outside the home), or (5) changing their cleaning habits (e.g., dusting more frequently). Some were dealing with their landlords, asking for environmental changes, such as new filters for heating/air systems or requesting pest control.

Only three mothers were visited by a staff member from the health department. There were two mothers who reported that the sanitarian was able to assist them in identifying asthma triggers within their homes. At one of these homes, the sanitarian treated for a roach infestation, and at the other home, the sanitarian was able to provide the mother with recommendations for cleaning products that did not include bleach. The third mother who received a home visit indicated that she did not receive a call back from the health department sanitarian and instead called the health department’s building inspection office herself. This mother received a home visit from a building inspector and indicated that the inspector was able to identify mold and roaches as triggers, however, this mother moved before the triggers could be corrected. There were three other mothers who moved before a home visit could be completed by the sanitarian. These mothers expressed concern that their living conditions were affecting their child’s health and felt that moving to a new home was their best available option for improving the health of their child. Mothers provided some insight into why they did not accept a home visit from the sanitarian. One mother stated she was fearful that a home visit could potentially upset her landlord and cause her to be evicted from her home, echoing a concern presented among health professionals for why mothers do not accept program referrals.

## 4. Discussion

The perceived positive impact of the program is consistent with other literature indicating multifaceted home-based environmental control programs with asthma education components are effective in improving asthma symptoms in children [2,6,18]. Health professionals viewed the program as a connecting mechanism, with the potential to connect parents with sanitarians at the local health department who could assess the home environment and recommend changes to improve it. However, a lack of feedback from the health department about whether follow-up occurred was viewed as a detriment. Health professionals at the children’s hospital were hoping for communication from sanitarians about whether families received visits and whether remediation of the home environment was occurring.

Mothers who completed interviews mentioned uncertainty over visits from the CHD sanitarians and often did not accept program referrals. Health professionals speculated that parents may have a mistrust or fear towards someone from the health department entering their home to conduct a home visit, indicating that the program may benefit from a more diverse group of personnel, who are already presently working within communities through existing community programs and partnerships, to complete home visits. Similar home intervention programs have utilized community health workers, health educators, nurses, sanitarians, and physicians to complete home visits as these professionals are well suited to work with low-income and ethnically diverse clients [11]. To this effect, Welker et al. [19] found that in successful asthma intervention programs, parents understood the potential health benefits of the interventions and the importance of maintaining changes in the home. This study adds to the literature, showing that support for changing housing conditions may be imperative for follow through for families residing in low-income neighborhoods. Understanding successes and understanding areas for improving home environments and the CLEAR program was used to inform hospital staff and the health department upon completion of this project. Continual parent feedback from a family advisory board may provide further insights into program successes and gaps in service provision and environmental needs in the home [10].

Recruiting and retaining clients is a common issue faced by asthma management programs [20]. Researchers in this study found this to be evident as many of the telephone numbers provided to the researcher by the health department were disconnected or the wrong telephone number. The Seattle-King County Asthma Program [11], and similar programs have found that utilizing a range of recruitment strategies is the most effective in reaching participants. Recruitment strategies from these programs include referral systems, participant incentives, small media advertising, participation in health fairs and other community events, word-of-mouth, and case-finding door-to-door surveys [11]. A need for further staff training at the hospital on the usages of the CLEAR program was presented in this study as an additional approach to improve program recruitment efforts. To this effect, huddles [21] were mentioned as an effective strategy among healthcare staff to receive and distribute crucial information, and perhaps an explanation of the program and how to better reach and assist families could be routinely addressed in huddle conversations.

This study had several limitations. The sample size of participants for this study was small, and those parents who participated had time constraints. However, redundancy in themes occurred. Additionally, this study occurred in only one setting, and assessing environmental health programs in different hospital settings may provide further information from different samples of health care professionals, further contributing to our understanding of children’s needs. Additionally, parent interviews were brief, which may have limited the insights that could be generated from this part of the study. The study was also limited by the fact that no fathers participated in interviews. Perspectives of fathers may have provided additional insight into some of their responsibilities that make asthma management in the home difficult for families. Providing incentives or conducting interviews at hospital visits for the child may be avenues for gaining more information from parents or caregivers. Health professionals who participated in interviews were scheduled through email at a time that best worked with their schedules. This could be why professionals were more inclined to answer all of the questions in their interviews. Interviewing other CHD personnel may have highlighted some of the challenges that the CHD faces in completing home visits and follow-up inspections.

## 5. Conclusions

The current study indicated that it will be necessary to find ways to reach and involve families in home visits, reduce fears of repercussions, and ensure caregivers feel comfortable accessing sanitarians at health departments. Mothers who did not wish to receive a home visit from the health department were able to use the education they received at the hospital (from health professionals) to correct some hazards in the home. Some mothers, who chose not to schedule a home visit with the sanitarian, had plans to move to a new home soon. It was not possible to ascertain if moves were solely to improve asthma management, and further information about this could be collected in additional research. It may be advantageous for health professionals to provide further explanation about how the program can help mothers during their child’s hospitalization, to ease parental concerns and make them feel more comfortable in accepting the services of the program. In summary, there were benefits to the program, in terms of addressing factors at the root of health disparities, providing information, and addressing environmental barriers that warrant its improvement and sustainability as a mechanism for connecting families to services and educating parents about how to improve the home environment to reduce asthma symptoms for children.

## Figures and Tables

**Table 1 ijerph-19-00172-t001:** Health professionals’ ideas for promoting sustainability and challenges to the sustainability of the CLEAR program.

Category	Theme	Number Endorsing Theme	Representative Quotes
Successful strategies for sustainability	The CLEAR program is a way to keep the home environment on the radar of health professionals	5	Interview 2: “We might know from the family their concern is mold or their concern is fire damage, but we may not. So I don’t know after it gets out of my hands as the physician what the next steps are.” Interview 4: “I think that having a program that allows families to receive assistance to help them do things to their home environment that they wouldn’t otherwise be able to do, and that’s keeping their child out of the hospital hopefully.”
	Patient feedback would show physicians that the program is working	4	Interview 2: “I think if you have positive patient outcomes that’s probably your best strategy for continued support of the program and continue to have physicians and nurse practitioners to want to use the program is when they see it benefits their patients.” Interview 8: “I don’t have the numbers about how many the health department actually goes out and talks to them, are people letting them in?” data
Challenges for sustainability	There are communication issues between parents and the health department that prevent them from connecting	5	Interview 1: “The challenges are making good contact with the guardian or the parent and even if contacting and inspection is made, having them be willing to coordinate a re-inspection and stay in the same unit is sometimes a challenge.” Interview 3: “We find families are hesitant to let somebody into their home. Whether they just don’t feel comfortable, or sometimes they are hiding something. The other thing is that sometimes it may not fully be explained to them why this program is here and why it’s implemented on certain homes.” Interview 4: “I think something that we are not always as good about as we should be is making sure that we have a reliable means to communicate with the family once they leave and it’s often sometimes as simple as we don’t have a reliable phone number, there phone number is changed, or their phone number gets turned off, or their voicemail box is full.” Interview 5: “I think there may be some concern about backlash from landlords if the health department comes in, you know they may have some fear around being evicted or are some negative consequences of them allowing this to happen and so I think those are the things that I think the families probably worry about.”
	Mistrust and fear make it difficult to connect with parents	7	Interview 2: “Families are often concerned about feeling like people are checking up on them.” Interview 3: “We find families are hesitant to let somebody into their home. Whether they just don’t feel comfortable, or sometimes they are hiding something. The other thing is that sometimes it may not fully be explained to them why this program is here and why it’s implemented on certain homes.” Interview 5: “I think there may be some concern about backlash from landlords if the health department comes in, you know they may have some fear around being evicted or are some negative consequences of them allowing this to happen and so I think those are the things that I think the families probably worry about.”

**Table 2 ijerph-19-00172-t002:** Health professionals’ perceptions of successful interventions of the CLEAR program.

Category	Theme	Number Endorsing Theme	Representative Quotes
Successful interventions of the CLEAR program	CLEAR can help address health disparities, get home services and home improvements for families in need	5	Interview 5: “Neighborhoods and the place they live can be a contributing factor to their asthma, I think the CLEAR program really helps us figure out what families need help and it allows us to address those needs sooner rather than later” Interview 7: “I think it essentially offers a service to help eradicate some of the household triggers that otherwise would not be available to any of our patients.”
	The program can help provide knowledge and resources to address mold and pests in the home data	6	Interview 5: “The CLEAR program is just a standardized way to get families the help they need for addressing things like mold and cockroaches, etc. or other environmental factors that might be contributing to their kids worsening asthma.”

**Table 3 ijerph-19-00172-t003:** Health professionals’ views of challenges in persuading families to accept and follow through with the referral process of the program.

Category	Theme	Number Endorsing Theme	Representative Quotes
Challenges in persuading families to accept and follow through with the referral process	Parents feel guilty or embarrassed that their home is impacting their child’s health	6	Interview 4: “I think that families feel responsibility and feel guilty about their child having health consequences because of that and feel embarrassed about that, and therefore may be hesitant to engage with people outside the hospital coming into their home and seeing things like that.” Interview 7: “People are just not interested in having other folks out to their homes, whether that’s the concern about being judged or what I can’t say.”

## Data Availability

Interview transcripts are available.

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
