# Peer review of "Evaluation of a Program to Reduce Home Environment Risks for Children with Asthma Residing in Urban Areas"

_ijerph, 2021, doi:10.3390/ijerph19010172_

Round 1

Reviewer 1 Report

I found this qualitative evaluation of the barriers to reduction in asthma related environmental factors to be well done and well presented. I have no substantial comments to suggest. Minorly, in the discussion in the paragraph starting at line 249: in addition to a diverse group of personnel I might also suggest  health workers/educators who are embedded in the comunity.  Either through partnerships with existing community groupds or through the development of community ambassadors. 

Reviewer 2 Report

This is a an interesting study aiming to perform an evaluation of an asthma home environmental intervention program by interviewing the stakeholders.

There are several minor issues that should be addressed in my opinion:

1. The sampling of the study population requires additional information.

2. The number and quality of interviewed people should be listed in the Results section (lines 106-115), not in the Methods section.

3. I suggest adding more details about the questions used for the interviews of health professionals.

4. The phrase on lines 71-74 is ambiguous - please reword/clarify.

5. Line 82: "who receive a received a referral to the program" - please correct

6. The phrase on line 117-119 is a bit ambiguous and should be reworded

7. Line 264: "to improve quality improvement..."

8. The Results section should be more structured around themes or areas of evaluation and duplicate information should be avoided (e.g. lines 171-173 and 202-204)

Reviewer 3 Report

This is a study on home environmental risks for children with asthma conducted on a small number of health care professionals and parents. Therefore, I do not believe that this study should be published before the authors add an additional number of participants, in order to conclude anything on this matter. Moreover, the manuscript itself has additional flaws, for instance, you should never start a sentence with a number, and you should use more references, especially those published in the last 2-3 years.

Round 2

Reviewer 3 Report

I believe this manuscript is now suitable for publication, although future studies should definitely include more participants.